# Efficient access to aliphatic esters by photocatalyzed alkoxycarbonylation of alkenes with alkyloxalyl chlorides

Jian-Qiang Chen [1✉], Xiaodong Tu[1], Qi Tang[1], Ke Li[1], Liang Xu[1], Siyu Wang[1], Mingjuan Ji[1], Zhiming Li [2✉] & Jie Wu [1,3,4✉]

Aliphatic esters are essential constituents of biologically active compounds and versatile chemical intermediates for the synthesis of drugs. However, their preparation from readily available olefins remains challenging. Here, we report a strategy to access aliphatic esters from olefins through a photocatalyzed alkoxycarbonylation reaction. Alkyloxalyl chlorides, generated in situ from the corresponding alcohols and oxalyl chloride, are engaged as alkoxycarbonyl radical fragments under photoredox catalysis. This transformation tolerates a broad scope of electron-rich and electron-deficient olefins and provides the corresponding β-chloro esters in good yields. Additionally, a formal β-selective alkene alkoxycarbonylation is developed. Moreover, a variety of oxindole-3-acetates and furoindolines are prepared in good to excellent yields. A more concise formal synthesis of (±)-physovenine is accomplished as well. With these strategies, a wide range of natural-product-derived olefins and alkyloxalyl chlorides are also successfully employed.

[1] School of Pharmaceutical and Materials Engineering & Institute for Advanced Studies, Taizhou University, Taizhou, China. [2] Department of Chemistry, Fudan University, Shanghai, China. [3] State Key Laboratory of Organometallic Chemistry, Shanghai Institute of Organic Chemistry, Chinese Academy of Sciences, Shanghai, China. [4] School of Chemistry and Chemical Engineering, Henan Normal University, Xinxiang, China. ✉email: chenjq@tzc.edu.cn; zmli@fudan.edu.cn; jie_wu@fudan.edu.cn

Aliphatic esters are highly valuable products and chemical intermediates. They are present in a broad range of important biologically active molecules (Fig. 1a)[1–3] and among the most versatile intermediates in the step-economical and orthogonal synthesis of aliphatic acids, aliphatic amides, aldehydes, ketones, and alcohols. Traditionally, synthesis of aliphatic esters relies on the esterification of carboxylic acids, anhydrides, or acyl chlorides with alcohols. These approaches need the pre-installation of a carboxyl group in the substrate. A complementary and more versatile alternative can use alkenes as starting materials, which are readily available and abundant petrochemical feedstock starting materials and synthetic intermediates[4]. Alkoxycarbonylation reactions of alkenes, developed by Reppe in the 1950s, is the most significant industrial process under transition-metal catalysis (Fig. 1b)[5]. Based on the palladium-catalyzed alkoxycarbonylation of ethylene, the current most advanced industrial process (Lucie Alpha process) to methyl propionate is used on an industrial scale to produce >300,000 tons of products annually[6]. Although transition-metal-catalyzed alkoxycarbonylation reactions are powerful tools to synthesize esters, these processes rely on the use of high pressure of CO, which often require specific equipment and safety precautions[7]. Additionally, application of these transformations is limited by the challenges associated with the regioselective alkoxycarbonylation of olefins as well as the harsh reaction conditions. Usually, a mixture of linear esters and branched esters is afforded[7]. Thus, direct catalytic and regioselective synthesis of aliphatic ester derivatives from unactivated olefins under mild conditions remains an unsolved challenge in modern synthetic chemistry. It is clear that an efficient approach is needed to address the above issues in the regioselective alkoxycarbonylation process.

We expected that a free radical-based method involving the addition of an alkoxycabonyl radical to olefin might be an attractive and alternative strategy. With this strategy, the alkoxycarbonylation of olefins will provide the desirable linear esters. In general, alkoxycarbonyl radicals are generated most from the corresponding diethyl azodicarboxylate[8], selenides[9], and xanthates[10]. Additionally, alkoxycarbonyl radicals can be formed from carbazates[11] and alkyl formates[12] by treatment with stoichiometric amounts of oxidants. Recently, it was reported that alkoxycarbonyl radicals could be produced by photoredox-catalyzed fragmentation of methyl *N*-phthalimidoyl oxalates (Fig. 1c)[13]. These existing methods for generating alkoxycarbonyl radicals from alcohols require multistep synthetic procedures. Moreover, most of the reported examples dealt with structurally simple alkoxycarbonyl radicals[12], while only a few reports exploiting complex alkoxycarbonyl radicals are described[14].

It is well known that single-electron reduction of aroyl and sulfonyl chlorides by photocatalyst would provide aroyl[15–17] and sulfonyl radicals[18], respectively. In this work, we envisioned that it might be possible to identify a strategy for the generation of alkoxycarbonyl radicals from the corresponding acyl chloride via photoredox catalysis (Fig. 1d). A photocatalytic strategy to introduce both the desired ester group and a versatile electrophile at the β-position of ester group would be quite useful for the preparation of significant compounds, due to the complementary reactivity of the esters. To the best of our knowledge, the selective alkoxycarbonylchlorination of alkenes leading to β-chloro esters from alkyloxalyl chlorides has not been explored yet.

## Results

**Reaction optimization.** To verify the feasibility of this free-radical alkoxycarbonylchlorination strategy, ethyl chloroformate (**2a'**) was used as the alkoxycarbonyl radical source. Unfortunately, none of the target product **3a** was obtained when a mixture of 4-vinyl-1,1'-biphenyl (**1a**), ethyl chloroformate (**2a'**), 2,6-lutidine and Ir(ppy)₃ in acetonitrile at 40 °C was irradiated with blue light-emitting diodes (LEDs) for 24 h (Table 1, entry 1). The failure of this result might be rationalized by the difficulty of promoting single-electron reduction of chloroformate because of its lower reduction potential. After careful investigation, we found that the readily available and inexpensive ethyl chlorooxoacetate (**2a**) [0.5kg/$227: supplier: Sigma-Aldrich and 0.5kg/$202: supplier: TCI] was an ideal alkoxycarbonyl radical precursor. This

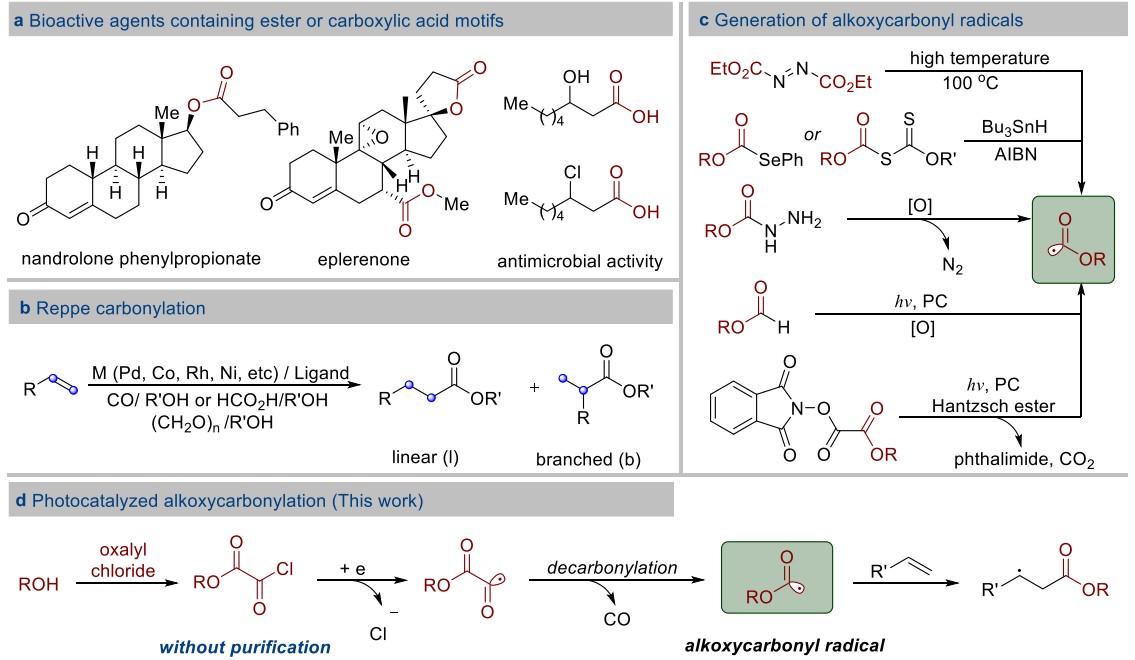

**Fig. 1 Bioactive carboxylic derivatives and approaches for their synthesis. a** Bioactive agents containing ester or carboxylic motifs. **b** Reppe carbonylation. **c** Generation of alkoxycarbonyl radicals. **d** This study.

**Table 1 Initial studies for the photoinduced reaction of 4-vinyl-1,1′-biphenyl 1a with ethyl chlorooxoacetate 2a[a].**

| Entry | PS | Solvent | T (°C) | Time | Yield (%) | 3a/6a[b] |
|-------|-----|---------|--------|------|-----------|----------|
| 1[c] | Ir(ppy)$_3$ | MeCN | 40 | 24 h | NR | — |
| 2 | Ir(ppy)$_3$ | MeCN | 40 | 24 h | 30 | 3:1 |
| 3 | 3DPA2FBN | MeCN | 40 | 24 h | 22 | >20:1 |
| 4 | Ru(bpy)$_3$Cl$_2$ | MeCN | 40 | 6 days | 42 | 14:1 |
| 5 | Ru(bpy)$_3$Cl$_2$ | MeCN | 60 | 72 h | 66 | 11:1 |
| 6 | Ru(bpy)$_3$Cl$_2$ | DMF | 60 | 72 h | 13 | 1:1.3 |
| 7 | Ru(bpy)$_3$Cl$_2$ | DCM | 60 | 72 h | 19 | 7:1 |
| 8[d] | Ru(bpy)$_3$Cl$_2$ | MeCN | 60 | 84 h | 85 | >20:1 |
| 9[d] | — | MeCN | 60 | 84 h | NR | — |
| 10[d,e] | Ru(bpy)$_3$Cl$_2$ | MeCN | 60 | 84 h | NR | — |

[a]Reaction conditions: 4-vinyl-1,1′-biphenyl **1a** (0.2 mmol), ethyl chlorooxoacetate **2a** (0.6 mmol), photocatalyst (0.004 mmol), 2,6-lutidine (0.6 mmol), solvent (4.0 mL), blue LEDs, under N$_2$ atmosphere. Conversion of **1a** and the yield of **3a** was determined by $^1$H NMR analysis using 1,3,5-trimethoxybenzene as an internal standard.
[b]Determined by $^1$H NMR analysis.
[c]Compound **2a′** was used instead of ethyl chlorooxoacetate **2a**.
[d]2,6-Lutidine (0.3 mmol).
[e]In the dark.

chlorooxoacetate might be readily reduced by the excited state of photocatalyst Ir(ppy)$_3$ to generate alkoxycarbonyl radical through CO extrusion. As shown in entry 2 (Table 1), the linear product **3a** was observed[19], accompanied by the appearance of by-product **6a** (10% yield) when ethyl chlorooxoacetate (**2a**) was used instead of ethyl chloroformate (**2a′**). This transformation proceeded with excellent regioselectivity and gave rise to the direct formation of linear ester **3a** (the branch isomer could not be found by $^1$H nuclear magnetic resonance (NMR)) under the operationally simple conditions. Additionally, examination of photosensitizers revealed that Ru(bpy)$_3$Cl$_2$ produced a better yield (entry 4); however, the reaction took much longer time for completion. To further improve the yield of alkoxycarbonylchlorination product of ethyl 3-([1,1′-biphenyl]-4-yl)-3-chloropropanoate (**3a**), the reaction temperature was increased, and the yield was slightly higher (66%, entry 5 in Table 1). Evaluation of different solvents showed that acetonitrile was the best choice in this transformation (entries 5–7). Notably, α,β-unsaturated ester compound **6a** was the major product when dimethylformamide (DMF) was used instead of acetonitrile. Interestingly, decreasing the amount of 2,6-lutidine led to a significantly improved yield and substrate (**1a**) was fully consumed (entry 8). As expected, visible light irradiation and photosensitizer were essential for this alkoxycarbonylchlorination reaction (entries 9–10).

**Substrate scope of activated alkenes**. With the optimized conditions in hands, the generality of this alkoxycarbonylchlorination reaction was then evaluated. As outlined in Fig. 2, a wide range of styrenes were efficiently workable in this protocol. For example, electron-neutral and electron-rich styrenes were all suitable substrates (**3b–3d**, 63–68% yield). This photocatalyzed alkoxycarbonylchlorination strategy was effective as well for electron-deficient styrenes, as demonstrated by the installation of fluoro, chloro, bromo, ester, and aldehyde groups (**3e–3i**, 41–75% yield). Furthermore, *para*-chloromethyl styrene, which could be further functionalized through nucleophilic substitution reaction, gave rise to the target product (**3j**) in good yield. Moreover, 1- and 2-vinylnaphthalenes were found to be competent substrates (**3k** and **3l**, 63 and 83% yield, respectively). The efficiency of this process was not impeded by *ortho*-, *meta*-methyl, or fluorine substitutions on the aromatic rings (**3m–3p**, 40–63% yield). A natural-product-derived styrene substrate was also coupled with high level of efficiency (**3q**, 62% yield). This result further demonstrated the potential of employing native functionality to access structural analogs and to provide further functionalization.

Having demonstrated the capacity of activated alkenes in this alkoxycarbonylchlorination reaction, we next investigated the conversion of various alkyloxalyl chlorides. As shown in Fig. 2, the yield was decreased to 38% (**3b** vs. **3r**) when commercially available methyl oxalyl chloride was used as the alkoxycarbonyl radical precursor in the reaction with styrene. By raising the equivalent of methyl oxalyl chloride, the yield of **3r** was increased from 38 to 54%. We applied this strategy to the derivatization of alcohol-containing biologically active molecules. By treatment of the corresponding alcohols with oxalyl chloride in dry dichloromethane (DCM), alkyloxalyl chlorides generated in situ were employed directly in the photoredox protocol after removal of the excess amount of oxalyl chloride and DCM by vacuum distillation[20]. Alkyloxalyl chlorides derived from primary and secondary alcohols were found to be successful in this alkoxycarbonylchlorination reaction. Chiral amino alcohol derivative reacted with styrene leading to product **3s** in 64% yield.

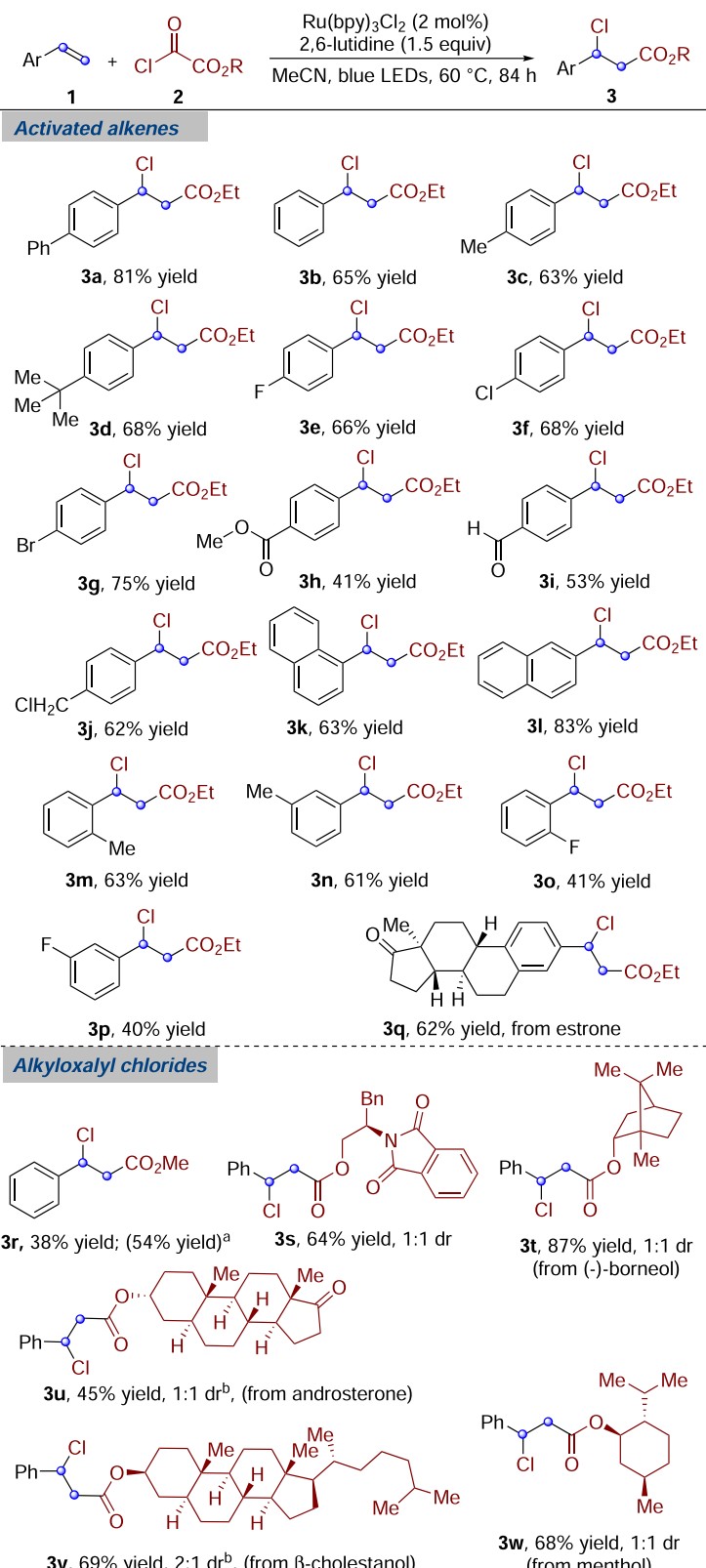

**Fig. 2 Alkoxycarbonylchlorination with activated alkenes.** Reaction conditions: activated alkene **1** (0.2 mmol), alkyloxalyl chloride **2** (0.6 mmol), Ru(bpy)$_3$Cl$_2$ (2 mol %), 2,6-lutidine (0.3 mmol), anhydrous MeCN (4.0 mL), blue LEDs, 60 °C, 84 h, under N$_2$ atmosphere. Isolated yields. The dr values were determined by $^1$H NMR analysis. [a]Alkyloxalyl chloride **2** (1.0 mmol). [b]Determined by HPLC analysis.

Additionally, this transformation was insensitive to steric hindrance around the site of alkoxycarbonyl radical (3t–3w, 45–87% yield). For example, alkyloxalyl chloride derived from (-)-borneol provided the desired product (3t) in 87% yield. Notably, chlorooxoacetates derived from other nature products, including androsterone (product 3u, 45% yield), β-cholestanol (product 3v, 69% yield) and menthol (product 3w, 68% yield), were workable as well. In general, reactions of secondary alkyl chlorooxoacetates gave better yields than those of primary alkyl chlorooxoacetates. This result might be rationalized by the slower decay rate of secondary alkoxycarbonyloxy radicals with alkyloxalyl chlorides ($\tau = 4.4$ µs for $i$-propylcarbonyloxy radical vs. $\tau = 2.4$ µs for ethylcarbonyloxy radical in $CCl_4$)[21,22].

**Reaction condition development.** Encouraged by the results described above, we subsequently explored the transformation of various unactivated alkenes (Supplementary Table 2). The reaction of non-conjugated alkene (but-3-en-1-ylbenzene 4a) failed to provide the corresponding product under the optimal conditions. However, the formation of desired β-chloro ester product 5a was observed when Ir(ppy)$_3$ was employed. A better result was obtained when the reaction was performed at 30 °C. By raising the equivalents of photosensitizer Ir(ppy)$_3$ and alkoxycabonyl radical source 2a, the yield of this reaction was increased from 44 to 74%.

**Substrate scope of unactivated alkenes.** On the basis of the above-optimized conditions, we next explored the reaction scope with diverse unactivated alkenes. As illustrated in Fig. 3, a wide range of unactivated alkenes coupled with chlorooxoacetate 2a efficiently, giving rise to the corresponding products. Transformation of long-chain α-olefins produced the desired compounds 5b–5e in reasonable yields (53–69%) with complete regiocontrol. Remarkably, hydrolysis of ethyl β-chlorooctanoate 5b promoted by HCl would provide an antibacterial compound[3]. Cetylates and stearates are privileged motifs encountered across the molecular sciences[23], particularly in food chemistry[23], medicinal chemistry[24], flavor, and fragrance industry[25]. Reaction of 1-pentadecene or 1-heptadecene would afford the addition product of ethyl β-chlorocetylate 5d or ethyl β-chlorostearate 5e in moderate yields, respectively. Moreover, a versatile electrophile at the β-position of ester would certainly accelerate the synthesis and discovery of bioactive molecules. More nucleophilic 1,1-disubstituted alkenes could participate smoothly in this transformation, affording the corresponding product 5f in 78% yield. It was found that a striking feature of this reaction was the exclusive formation of β-chloro esters without any undesired rearranged products, even in cases where benzyl, cyclohexyl, and *tert*-butyl were present in the α-position of double bond. Aside from various carbon scaffolds, some functional groups were found to be tolerated under the conditions, such as esters (5j and 5k), ketones (5l and 5m), and amide (5o). Reaction of tri-substituted alkene derived from cholesterol produced the corresponding product 5n in an unoptimized 21% yield, along with 40% of unreacted starting material recovered. Another aspect worth mentioning here was the excellent diastereoselectivity (>20:1 dr) and regioselectivity of this transformation. Furthermore, electron-deficient olefin was entirely converted into the expected β-chloro product 5o. In all cases, only one regioisomer was obtained, making this reaction fully regioselective.

Since 1-tetralone moiety is widely found in the core structure of natural products[26], organic synthetic intermediates[27], and bioactive compounds[28,29], much attention has focused on the synthesis of 1-tetralone derivatives. Therefore, the above method was applied to the preparation of 1-tetralone derivatives. Several

1-tetralone derivatives (5') were prepared in moderate yields through a radical alkoxycarbonylation/cyclization with 1-aryl-pent-4-en-1-ones under the same reaction conditions. This convenient method described above will be potentially useful for the synthesis of bioactive compounds containing 1-tetralone moiety.

**Synthesis of α,β-unsaturated esters.** Late-stage carbon–hydrogen bond functionalization of pharmacologically active compounds is a remarkable strategy for the discovery of functional compounds because it avoids laborious de novo construction of analogs, increases the efficiency of structure–activity relationship investigation, and provides candidates that might have never been explored[30]. Since β-chloro esters generated from direct alkoxycarbonylchlorination could undergo HCl elimination upon workup leading to α,β-unsaturated esters[31], we envisioned that this formal β-selective alkenyl C–H alkoxycarbonylation could be accomplished and quite appealing. Additionally, α,β-unsaturated esters are key components of synthetic building blocks, pharmaceuticals, and natural products[32–34]. Thus, synthesis of α,β-unsaturated esters remains an actual interesting task in the context of development of improved synthetic methodologies[35]. As illustrated in Fig. 4, by treatment of alkenes and ethyl chlorooxoacetate 2a under the optimal conditions, followed by elimination in the presence of excess amount of 1,8-diazabicyclo[5.4.0]undec-7-ene (DBU) at 25 °C for 30 min, α,β-unsaturated esters 6 were obtained with high levels of chemoselectivity, regioselectivity, and stereoselectivity. A wide range of olefins could participate in this transformation, affording the corresponding α,β-unsaturated esters. Various electron-neutral, electron-rich, and electron-poor styrenes were viable substrates (6a–6j, 47–75% yield). 1,1-Disubstituted aryl alkenes reacted with ethyl chlorooxoacetate 2a giving rise to the corresponding trisubstituted α,β-unsaturated esters in moderate yields (6k–6l, 54–59% yield). Reaction of 3,4,5-trimethoxystyrene could produce the desired cinnamic ester derivative 6m. To demonstrate the amenability of this alkoxycarbonylation process to late-stage application, bexarotene[36] analog was subjected to the standard conditions and the corresponding alkoxycarbonylation product 6n was obtained in 57% yield. Additionally, unactivated alkenes could be workable as well, affording the desired products (6o and 6p) in moderate yields.

**Synthesis of oxindole-3-acetates.** We next explored the generality of the decarbonylative alkoxycarbonylation/cyclization protocol. As shown in Fig. 5, it was found that a broad range of N-arylacrylamide derivatives and alkyloxalyl chlorides were suitable substrates in this transformation. A number of N-arylacrylamides bearing both electron-rich and electron-poor substituents in the aromatic ring underwent alkoxycarbonylation/cyclization with ethyl chlorooxoacetate 2a leading to the corresponding products in good-to-excellent yields. Notably, reaction of antetrahydroquinoline or tetrahydrobenz-azepine derivative afforded the desired tricyclic product 8e or 8f in 91 and 90% yield, respectively. Furthermore, N-arylacrylamide with a naphthalene substituent on the nitrogen reacted with ethyl chlorooxoacetate 2a smoothly, providing the corresponding oxindole-3-acetate 8l in 83% yield. N-arylacrylamides with ethyl-, isopropyl-, and benzyl-protecting groups on the nitrogen atom proceeded well, affording the substituted products in good yields (8m–8o, 72–80% yield). Additionally, acrylamide with a benzyl group at the α-position could convert into the desired product 8p in 91% yield. To demonstrate the practicability of this photoredox process, a gram-scale experiment was carried out, which provided the corresponding oxindole-3-acetate 8a in 77% yield (1.53 g). However, transformations of unprotected N-arylacrylamide,

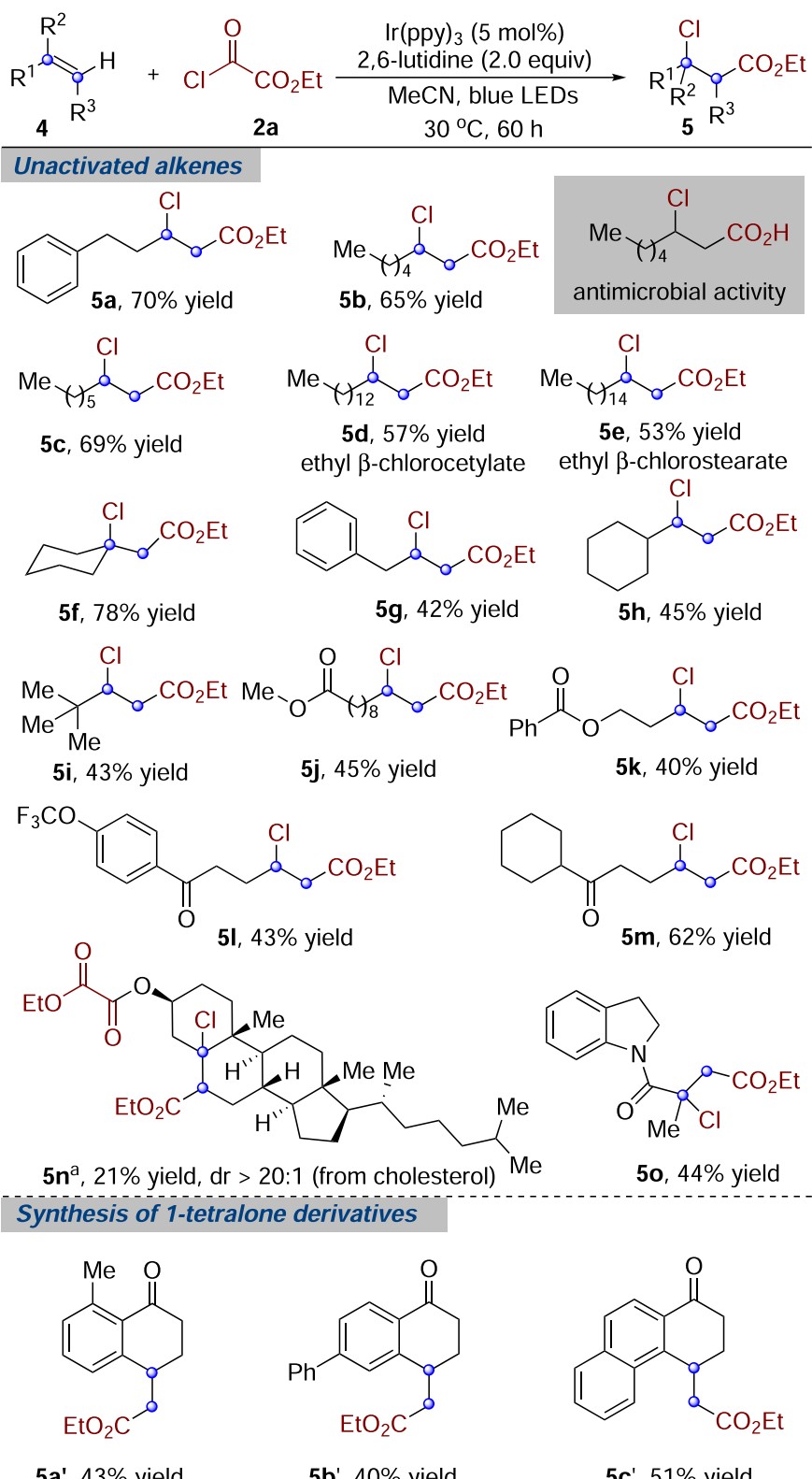

**Fig. 3 Alkoxycarbonylchlorination with unactivated alkenes.** Reaction conditions: unactivated alkene **4** (0.2 mmol), ethyl chlorooxoacetate **2a** (1.6 mmol), Ir(ppy)$_3$ (5 mol %), 2,6-lutidine (0.4 mmol), anhydrous MeCN (4.0 mL), blue LEDs, 30 °C, 60 h, under N$_2$ atmosphere. Isolated yields. [a]Reaction time: 84 h. The dr value was determined by $^1$H NMR analysis.

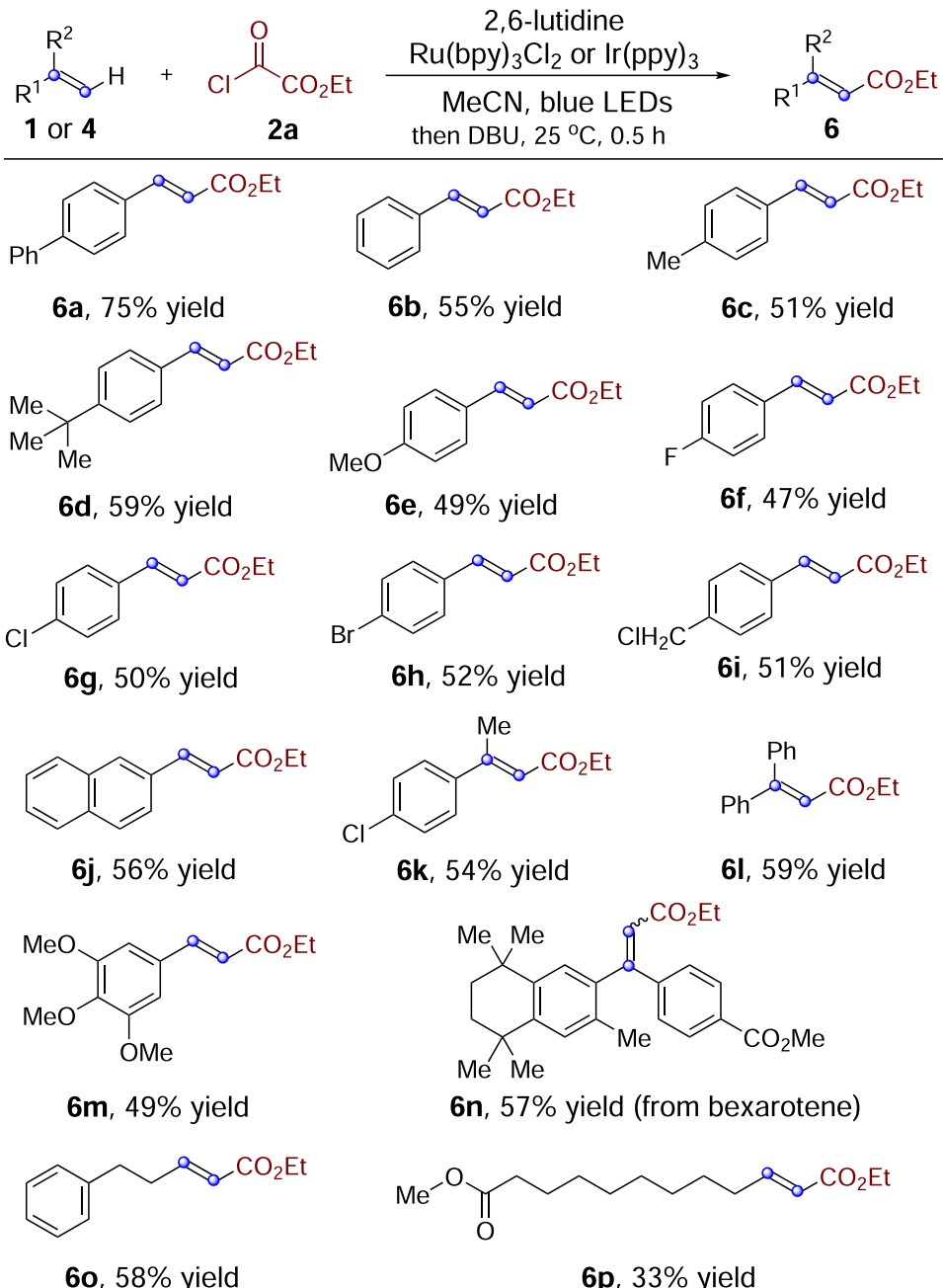

**Fig. 4 Synthesis of α,β-unsaturated esters.** Isolated yields based on alkene **1** or **4**.

mono-substituted olefin (R³ = H), and indoline derivative were not successful.

Subsequently, reactions of various alkyloxalyl chlorides were investigated under the standard conditions. The yield of alkoxycarbonylation/cyclization product was slightly diminished (**8q** vs. **8a**) when methyl oxalyl chloride was used instead of ethyl chlorooxoacetate **2a**. Alkyloxalyl chlorides derived from primary alcohols were found to be successful in this conversion. For example, reaction of alkyloxalyl chloride derived from 4-phenyl-1-butanol provided the desired product in quantitative yield (**8r**, 99% yield). The long-chain alkyloxalyl chloride derivative was also an effective alkoxycarbonyl radical source, and the corresponding product **8s** was furnished in 85% yield. To further demonstrate the advantage of this conversion, we applied this strategy to the derivatization of alcohol-containing biologically active molecules. A series of chiral secondary alcohols were

examined, and the representative examples are shown in Fig. 5. It is noteworthy that the corresponding chiral secondary alkoxycarbonyl radical intermediates could readily convert into the desired products without any decarboxylated products (**8t–8y**, 58–84% yields). Moreover, the reaction was insensitive to steric hindrance around the site of alkoxycarbonyl radicals (**8t–8v**, 61–84% yield). The transformations proceeded efficiently as well when β-cholestanol (product **8w**, 81% yield), androsterone (product **8x**, 81% yield), and cholesterol-derived (product **8y**, 58% yield) alkyloxalyl chlorides were used. Excellent diastereoselectivity (>20:1 dr) was observed for β-cholestanol substrate. We speculated that the unique three-dimensional structure of β-cholestanol might influence the following cyclization reaction. Chiral amino alcohol derivatives could also be applied to the synthesis of oxindole-3-acetates (**8z–8ad**, 67–82% yield). These experiments demonstrated that this strategy was compatible with

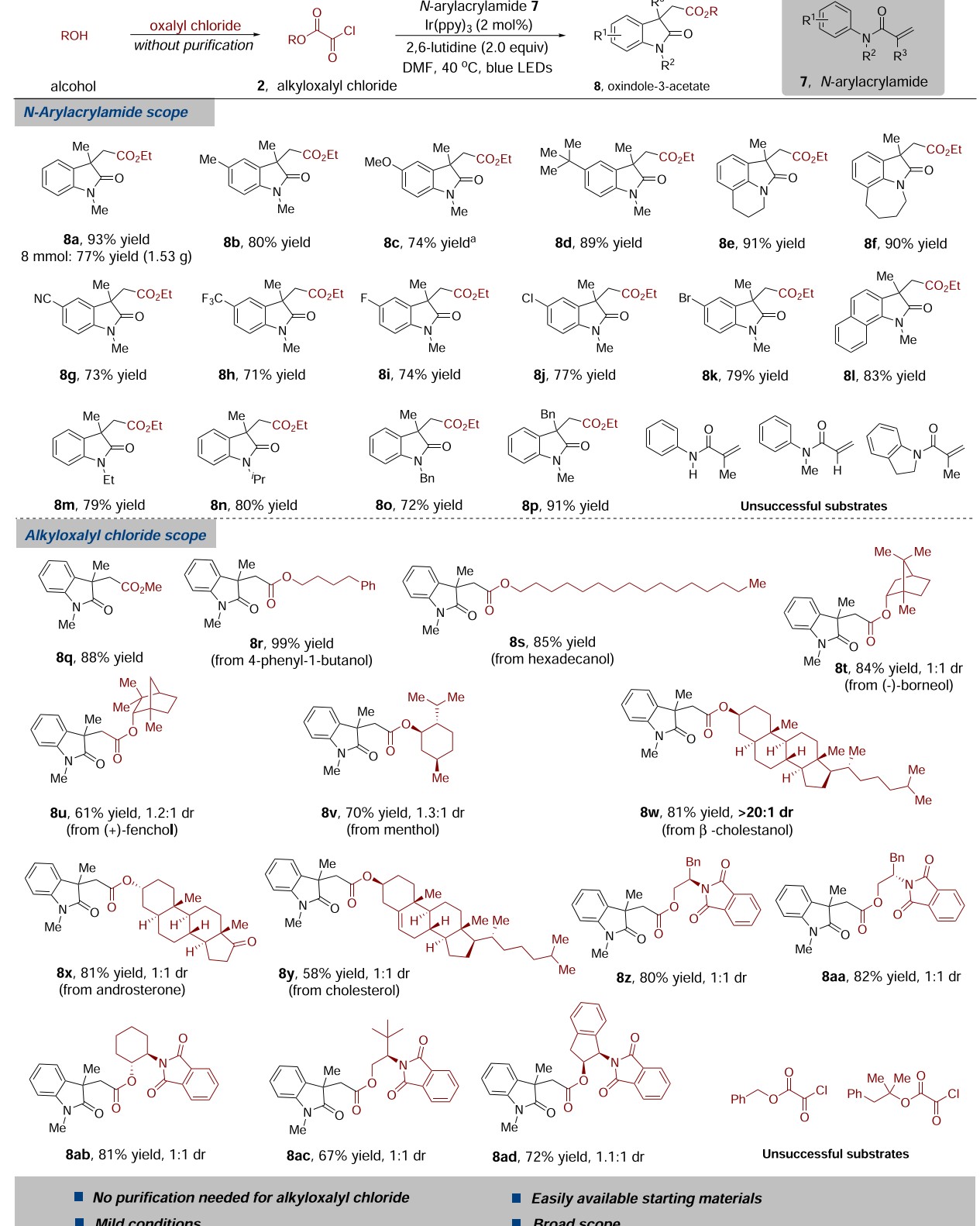

**Fig. 5 From alcohols to oxindole-3-acetates: substrate scope of the alkoxycarbonylation/cyclization reaction.** Reaction conditions: *N*-arylacrylamide **7** (0.2 mmol), alkyloxalyl chloride **2** (0.6 mmol), Ir(ppy)₃ (2 mol %), 2,6-lutidine (0.4 mmol), anhydrous DMF (4.0 mL), blue LEDs, 40 °C, 24 h, under N₂ atmosphere. Yield of isolated product. The dr values were determined by ¹H NMR analysis. ᵃReaction time: 48 h.

the functionalization of biologically active molecules bearing polar functional groups (**8x** and **8z–8ad**). However, reactions of benzyl and tertiary alcohol derivatives failed to provide the corresponding products.

**Synthesis of furoindolines**. Furoindoline moiety is broadly found in the core structure of biologically active compounds and natural products[37–44]. Considering that oxindole-3-acetates are versatile building blocks for constructing heterocycle-fused indolines[45,46], we decided to apply the above method to the preparation of furoindoline derivatives. As shown in Fig. 6, by treatment of *N*-arylacrylamides **7** and ethyl chlorooxoacetate **2a** under the optimal conditions, followed by reduction with LiAlH$_4$ at 0 °C, furoindoline **9** was obtained as expected with excellent diastereoselectivities[45]. This route was highly efficient, and a range of furoindoline derivatives was readily produced in only two steps from simple precursors. Additionally, *N*-(4-methoxyphenyl)-*N*-methylmethacrylamide reacted with ethyl chlorooxoacetate **2a** under the standard conditions, giving rise to the desired tricyclic furoindoline **9m**, which could be readily converted into bioactive alkaloid physovenine in two steps[37,38]. In

contrast to previous reports for the synthesis of physovenine, including Sharpless epoxidation[39], Grignard reaction[40], Diels–Alder reaction[41], catalytic asymmetric Heck reaction[42], intramolecular Michael addition[43], and [3,3]-sigmatropic rearrangement[44], this method was much better from the viewpoint of atom- and step-economy.

**Synthetic application**. We next applied this methodology to the concise synthesis of expensive dihydronaphthalene derivative [5g/¥37,438; supplier: Biofount], which is an important precursor for a wide range of biorelevant molecules[47]. In contrast to the conventional routes[48,49], our method not only decreased the step count but also simplified the operation greatly (Fig. 7a). Furthermore, this strategy was utilized in the formal synthesis of marketed drug ozagrel, which was an antiplatelet drug and marketed in Japan in 1989[50–52]. As shown in Fig. 5, this alkoxycarbonylation reaction enabled us to access key compound **6i**, which could be subsequently subjected to the substitution reaction with imidazole providing product **10i** in 70% yield. An additional hydrolysis of compound **10i** would afford ozagrel (Fig. 7b)[52]. Overman demonstrated that methoxycarbonyl radical could react with electron-deficient olefins

**Fig. 6 Synthesis of furoindoline derivatives.** Isolated yields based on *N*-arylacrylamide **7**.

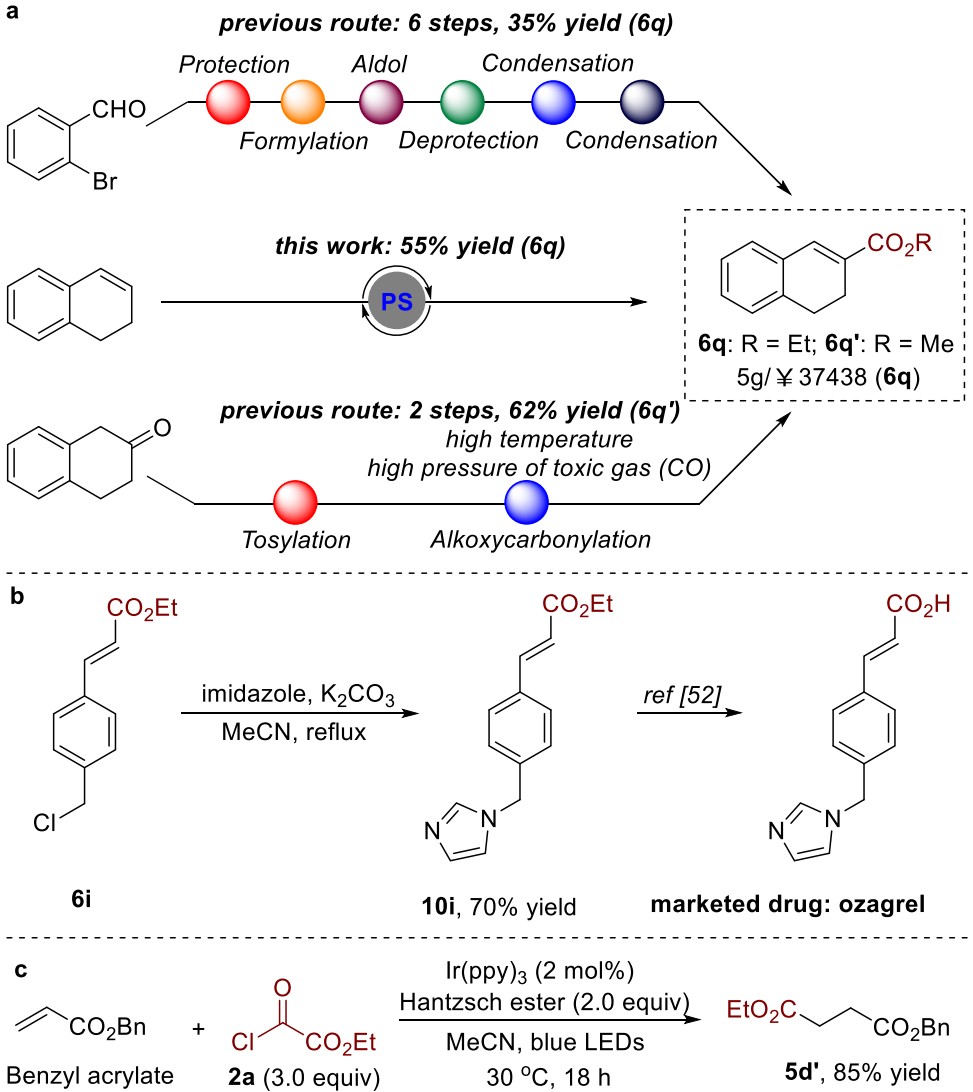

**Fig. 7 Synthetic application. a** Synthesis of expensive dihydronaphthalene derivative. **b** Formal synthesis of marketed drug ozagrel. **c** Synthesis of 1,4-dicarbonyl compound.

to afford 1,4-dicarbonyl compounds[13]. We next applied our methodology to this transformation. As expected, coupling of ethyl chlorooxoacetate **2a** (3.0 equiv) with benzyl acrylate in the presence of Ir(ppy)₃ (2 mol %) and diethyl 1,4-dihydro-2,6-dimethyl-3,5-pyridinedicar-boxylate (2.0 equiv) in MeCN with blue LED irradiation at 30 °C gave rise to 1,4-dicarbonyl compound **5d'** in 85% yield (Fig. 7c).

**Mechanistic studies**. To elucidate the possible reaction mechanism, a radical clock experiment with (1-cyclopropylvinyl) benzene **1x** as the substrate was preformed, and a ring-opening product was obtained as expected (Fig. 8a). This result clearly demonstrated the involvement of an alkoxycarbonyl radical. Additionally, a radical trapping experiment with the addition of 2,2,6,6-tetramethyl-1-piperidyloxy (TEMPO) showed that the related alkoxycarbonyl-TEMPO product was confirmed through gas chromatography–mass spectrometry (for details, see Supplementary Information). It is noteworthy that the presence of 2,6-lutidine was significant for the success of these transformations. As mentioned above, both photosensitizer and visible light were necessary for this alkoxycarbonylation process (Table 1, entries 9 and 10). Thus, these results could rule out the ethyl chlorooxoacetate activation by the formation of electron–donor–acceptor

complex[53]. As shown in Fig. 9b, the presence of 2,6-lutidine enhanced the reduction potential of ethyl chlorooxoacetate **2a** from $E_p$ (**2a**) = −1.23 V vs. Ag/AgCl to $E_p$ (**2a** + 2,6-lutidine) = −1.09 V vs. Ag/AgCl. ¹H NMR studies showed a critical downfield shift of 2,6-lutidine protons after the addition of ethyl chlorooxoacetate **2a** (for details, see Supplementary Fig. 4), indicating the generation of acyl pyridinium salt **I-2a**. The possible structure of **I-2a** is shown in Fig. 9. These results implied that there was no π conjugation between the aromatic ring and acyl plane. Stern–Volmer experiments revealed that ethyl chlorooxoacetate **2a** and acyl pyridinium salt **I-2a** could quench the excited photocatalyst, but alkene and 2,6-lutidine could not. The quenching constant for *Ir(ppy)₃ with ethyl chlorooxoacetate **2a** was determined as $k_q = 1.70 \times 10^8 \, M^{-1} \, s^{-1}$, which was 1.38 times faster than the reaction of *Ir(ppy)₃ with **I-2a**. We speculated that the unique three-dimensional structure of **I-2a** might influence the single electron transfer from the excited photocatalyst to **I-2a**. Notably, the addition of Cl⁻ would slightly improve the yield of **3a** from 3 to 20% (Fig. 8d). This result suggested the feasibility of carbon cation pathway. The quantum yields of these reactions were determined to be 1.11, 1.33, and 9.98, showing that the extended radical-chain reactions were possible. Moreover, bond dissociation energy of C–H bond could reflect the stability of

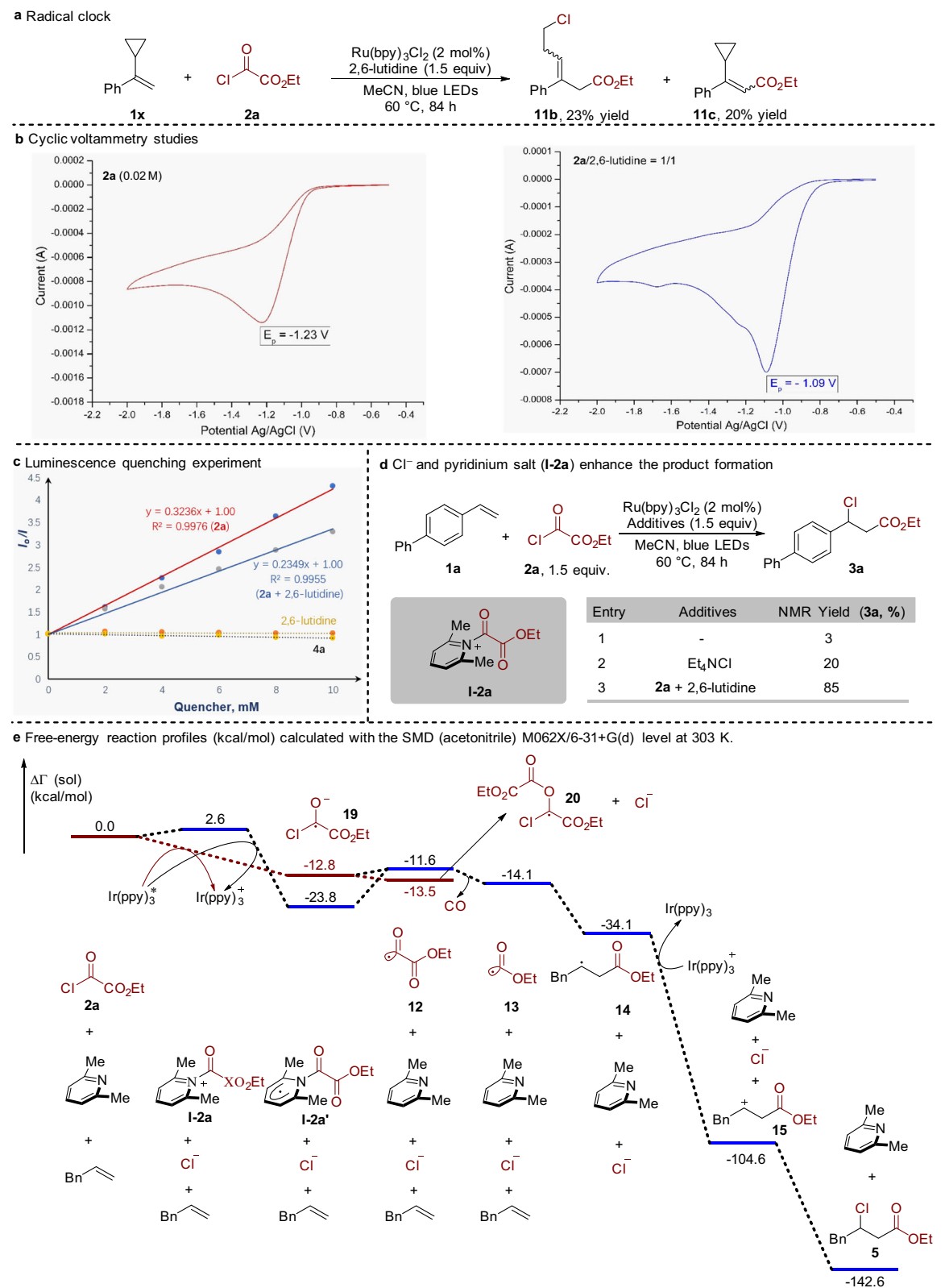

**Fig. 8 Mechanistic studies. a** Radical clock experiment. **b** Cyclic voltammetric studies. **c** Luminescence quenching experiment. **d** Cl⁻ and pyridinium salt (**I-2a**) enhance the product formation. **e** Free-energy reaction profiles (kcal mol⁻¹) calculated with the SMD (acetonitrile) M062X/6-31 + G(d) level at 303 K.

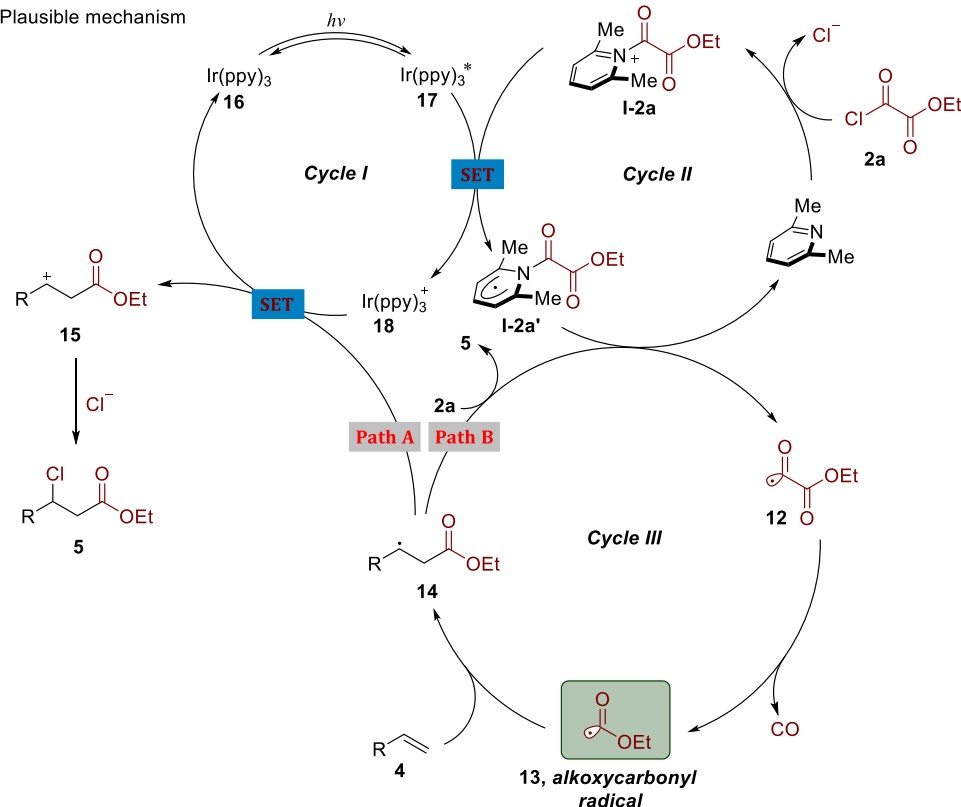

Plausible mechanism

**Fig. 9 Plausible mechanism for photocatalyzed alkoxycarbonylchlorination.** Cycle I: photocatalytic cycle. Cycle II: 2,6-lutidine-initiated catalytic cycle with compound **2a**. Cycle III: chain pathway for the generation of alkoxycarbonyl radical **13** via decarbonylation.

carbon radicals. The C–H bond dissociation energy of MeOC(O)–H and tBu–H are 95.4[54] and 95.6 kcal mol−1 [55], respectively, and the stability of alkoxycarbonyl radical might be similar to the tertiary carbon radical.

In order to shed more light on the reaction mechanism, especially the role of 2,6-lutidine, density functional theory (DFT) calculations were carried out with the Gaussian 09 software package[56–58]. The calculation details were provided in the Supplementary Information. As shown in Fig. 9e, the whole reaction was thermodynamically favorable and the driving force for this transformation was the extrusion of carbon monoxide, the generation of stable alkoxycarbonyl radical **13**, and subsequent reaction with alkene to afford alkyl radical **14**. Concerning the formation of key intermediate **12**, though the generation of acyl pyridinium salt **I-2a** was a little bit endothermic, the formation of **I-2a'** via reduction of **I-2a** by highly reducing species *Ir(ppy)₃ was more thermodynamically favored compared to the direct formation of **19** without partition of 2,6-lutidine. Additionally, anion radical **19** could react with compound **2a** to deliver the undesired alkyl radical **20**. Thus, 2,6-lutidine might facilitate the transformation relative to no participation of 2,6-lutidine. Moreover, Stern–Volmer experiments have demonstrated that the step to generate anion radical **19** is kinetically favorable. Hence, raising the equivalent of 2,6-lutidine would facilitate the generation of the **I-2a'** and suppress the formation of anion radical **19**. As shown in Supplementary Fig. 15, DFT calculations were also applied to study the side reaction of ethoxycarbonyl radical and ethyl chlorooxoacetate to generate radical **12** and ethyl carbonochloridate. The reaction free energy was 0.6 kcal mol−1, while the reaction free energy of methoxycarbonyl radical and methyloxalyl chloride was only 0.1 kcal mol−1. According to Arrhenius equation, the side reaction of the latter was more than the former. This might

contribute to the low yield of methyloxalyl chloride compared to ethyloxalyl chloride (**3b** vs. **3r**).

On the basis of the above experimental results and previous reports[59], a plausible reaction mechanism is proposed in Fig. 9. Excitation of photosensitizer Ir(ppy)₃ **16** with blue light would generate a long-lived excited *Ir(ppy)₃ ($\tau = 1.9 \mu s$)[59]. Meanwhile, 2,6-lutidine would condense with ethyl chlorooxoacetate **2a** to form acyl pyridinium salt **I-2a**. The highly reducing species *Ir(ppy)₃ **17** ($E_{1/2}$ [Ir(ppy)₃⁺/*Ir(ppy)₃] = −1.73 V vs. saturated calomel electrode) would reduce intermediate **I-2a** leading to radical intermediate **I-2a'** and Ir(ppy)₃⁺ **18**. The radical intermediate **I-2a'** would undergo C–N homolysis to form acyl radical intermediate **12** with the elimination of 2,6-lutidine[60,61]. Acyl radical **12** would then undergo decarbonylation, giving rise to alkoxycarbonyl radical **13**[62,63]. The driving force for this transformation is the extrusion of carbon monoxide and the generation of stable alkoxycarbonyl radical. This ambiphilic radical **13** could react with an alkene to afford alkyl radical **14**, which would be readily oxidized by Ir(ppy)₃⁺ **18** to regenerate ground-state photocatalyst **16** and carbocation intermediate **15**. This carbocation intermediate **15** would be attacked by chloride anion leading to the desirable β-chloro ester **5** (path A). A competitive chain pathway could not be excluded (path B). Alkyl radical **14** would abstract chlorine atom from ethyl chlorooxoacetate **2a** to produce the target product **5** and regenerate acyl radical intermediate **12**.

## Discussion

In summary, we have described the generation and application of alkoxycarbonyl radicals under photoredox catalysis from alkyloxalyl chlorides, generated in situ from the corresponding alcohols and oxalyl chloride. This photocatalytic strategy to introduce both the desired ester group and a versatile electrophile at the β-

position of ester group is quite useful for the preparation of significant compounds, due to its complementary reactivity. Additionally, a formal β-selective alkene alkoxycarbonylation is described. With this approach, a variety of oxindole-3-acetates and furoindolines are prepared in good-to-excellent yields through alkoxycarbonylation/cyclization with *N*-arylacrylamides under mild conditions. Additionally, this strategy can be compatible with the derivatization of alcohol-containing biologically active molecules. A more concise formal synthesis of (±)-physovenine is accomplished as well. All these results further demonstrate the potential of employing native functionality to access structural analogs and to provide the late-stage functionalization.

## Methods

**General procedure for the synthesis of compound 3**. Substrate **1** (0.2 mmol), alkyloxyoxalyl chloride **2** (0.6 mmol), and 2,6-lutidine (32.1 mg, 0.3 mmol) were added to a solution of Ru(bpy)$_3$Cl$_2$ (3.0 mg, 2 mol %) in dry MeCN (4.0 mL) at 25 °C. The heterogeneous mixture was degassed by three cycles of freeze–pump–thaw and then placed in the irradiation apparatus equipped with blue LEDs. The resulting mixture was stirred at 60 °C for 84 h. Upon completion of the reaction, the mixture was diluted with ethyl acetate (30 mL), washed with brine (10 × 3 mL), and dried with Na$_2$SO$_4$. After evaporation of the solvent, the crude product was purified by column chromatography on silica gel to afford the desired product **3**.

**General procedure for the synthesis of compound 5 or 5′**. Substrate **4** (0.2 mmol), alkyloxyoxalyl chloride **2a** (218.4 mg, 1.6 mmol), and 2,6-lutidine (42.8 mg, 0.4 mmol) were added to a solution of Ir(ppy)$_3$ (6.54 mg, 5 mol %) in dry MeCN (4.0 mL) at 25 °C. The heterogeneous mixture was degassed by three cycles of freeze–pump–thaw and then placed in the irradiation apparatus equipped with blue LEDs. The resulting mixture was stirred at 30 °C for 60 h. Upon completion of the reaction, the mixture was diluted with ethyl acetate (30 mL), washed with brine (10 × 3 mL), and dried with Na$_2$SO$_4$. The solvent was evaporated, and the crude product was purified by column chromatography on silica gel to afford the desired product **5** or **5'**.

**General procedure for the synthesis of compound 6**. Substrate **1** (0.2 mmol), chlorooxoacetate **2a** (81.9 mg, 0.6 mmol), and 2,6-lutidine (32.1 mg, 0.3 mmol) were added to a solution of Ru(bpy)$_3$Cl$_2$ (3.0 mg, 2 mol %) in dry MeCN (4.0 mL) at 25 °C. The heterogeneous mixture was degassed by three cycles of freeze–pump–thaw and then placed in the irradiation apparatus equipped with blue LEDs. The resulting mixture was stirred at 60 °C for 84 h. Upon completion of the reaction, 1,8-diazabicyclo[5.4.0]undec-7-ene (152.2 mg, 1.0 mmol) was added and the mixture was stirred at 25 °C for 0.5 h. The mixture was diluted with ethyl acetate (30 mL), washed with brine (10 × 3 mL), and dried with Na$_2$SO$_4$. The solvent was then evaporated, and the crude product was purified by column chromatography on silica gel to afford the desired product **6** (**6a–6n** and **6q**).

Substrate **4** (0.2 mmol), alkyloxyoxalyl chloride **2a** (218.4 mg, 1.6 mmol), and 2,6-lutidine (42.8 mg, 0.4 mmol) were added to a solution of Ir(ppy)$_3$ (6.54 mg, 5 mol %) in dry MeCN (4.0 mL) at 25 °C. The heterogeneous mixture was degassed by three cycles of freeze–pump–thaw and then placed in the irradiation apparatus equipped with blue LEDs. The resulting mixture was stirred at 30 °C for 60 h. Upon completion of the reaction, the mixture was diluted with ethyl acetate (30 mL), washed with brine (10 × 3 mL), and dried with Na$_2$SO$_4$. After evaporation of the solvent, the crude product was dissolved in tetrahydrofuran (THF; 4 mL) and DBU (152.2 mg, 1.0 mmol) was added. The reaction mixture was stirred at 25 °C for 0.5 h. Then the mixture was diluted with ethyl acetate (30 mL), washed with brine (10 × 3 mL), and dried with Na$_2$SO$_4$. After evaporation of the solvent, the crude product was purified by column chromatography on silica gel to afford the desired product **6** (**6o** and **6p**).

**General procedure for the synthesis of compound 8**. Substrate **7** (0.2 mmol), alkyloxyoxalyl chloride **2** (0.6 mmol), and 2,6-lutidine (42.8 mg, 0.4 mmol) were added to a solution of Ir(ppy)$_3$ (2.62 mg, 2 mol %) in dry DMF (4.0 mL) at 25 °C. The heterogeneous mixture was degassed by three cycles of freeze–pump–thaw and then placed in the irradiation apparatus equipped with blue LEDs. The resulting mixture was stirred at 40 °C until the starting material was completely consumed as monitored by thin-layer chromatography (TLC). Upon completion of the reaction, the mixture was diluted with ethyl acetate (30 mL), washed with brine (10 × 3 mL), and dried with Na$_2$SO$_4$. After evaporation of the solvent, the crude product was purified by column chromatography on silica gel to afford the desired product **8**.

**General procedure for the synthesis of compound 9**. Substrate **7** (0.2 mmol), ethyl chlorooxoacetate **2a** (81.9 mg, 0.6 mmol) and 2,6-lutidine (42.8 mg, 0.4 mmol) were added to a solution of Ir(ppy)$_3$ (2.62 mg, 2 mol %) in dry DMF (4.0 mL) at 25 °C. The heterogeneous mixture was degassed by three cycles of freeze–pump–thaw and then placed in the irradiation apparatus equipped with blue LEDs. The resulting mixture was stirred at 40 °C until the starting material was completely consumed as monitored by TLC. Upon completion of the reaction, the mixture was diluted with ethyl acetate (30 mL), washed with brine (10 × 3 mL), and dried with Na$_2$SO$_4$. After evaporation of the solvent, the crude product was used in the following step without further purification. To a solution of crude product in THF (4.0 mL) at 0 °C was added LiAlH$_4$ (38 mg, 1.0 mmol) in small portions under nitrogen atmosphere. The reaction mixture was stirred at 0 °C for 2 h, and then the reaction was quenched with the addition of brine (15 mL) and diluted with EtOAc (30 mL). The combined organic layers were washed with brine (3 × 10 mL), dried over Na$_2$SO$_4$, filtered, and concentrated under reduced pressure. The resulting residue was purified by flash column chromatography with gradient eluents (*n*-hexane/ethyl acetate = 20/1) to provide compound **9**.

## Data availability

The data that support the findings of this study are available within the paper and its Supplementary Information files. Raw data are available from the corresponding author on reasonable request. Materials and methods, experimental procedures, characterization data, $^1$H and $^{13}$C NMR spectra, and mass spectrometric data are available in the Supplementary Information.

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

## Acknowledgements

Financial support from National Natural Science Foundation of China (No. 21871053), the Leading Innovative and Entrepreneur Team Introduction Program of Zhejiang (No. 2019R01005), and the Open Research Fund of School of Chemistry and Chemical Engineering, Henan Normal University (2020ZD04) is gratefully acknowledged.

## Author contributions

J.-Q.C. and J.W. conceived and supervised the whole project and wrote the paper with input from all authors. J.-Q.C., X.T., and J.W. designed and discussed the experiments. J.-Q.C., X.T., Q.T., K.L., L.X., S.W., and M.J. performed and analyzed the experiments. Z.L. performed the DFT calculations.

## Competing interests

The authors declare no competing interests.
