## [Peer Review File · Nature Communications]

Efficient access to aliphatic esters by photocatalyzed alkoxy-carbonylation of alkenes with alkyloxalyl chloridesREVIEWER COMMENTS

Reviewer #1 (Remarks to the Author):

Wu et al. reported a photoredox-catalyzed alkoxyacylation of alkenes, which provides a synthetic route for rapid access to aliphatic esters. This work has a very rich content that contains a series of reactions, including alkoxyacylation chlorination of activated and unactivated alkenes, synthesis of *a,b*-unsaturated esters, and alkoxyacylation/cyclization reaction for the synthesis of oxindoles and furoindoline derivatives. Each of these reactions could be a paper in itself. While the generation of alkoxyacylation radical under photocatalytic conditions is known using *N*-phthalimidoyl oxalates (ref. 12), this work takes another approach employing chlorooxalacetate derivatives as alkoxyacylation radical precursors. The authors also demonstrated the synthetic applications of the alkoxyacylation reaction to streamline the synthesis of several drug molecules and expansive synthetic intermediates. Thus, this work is significant and will likely be of interest to others in the community. However, the reaction time is very long (84 h!), which may negatively impact its synthetic applications. In addition, this paper is missing (i) many explanations and rationales for different observations, (ii) mechanistic studies and understandings (see comments below). Nevertheless, if the authors could address the following comments and issues, this manuscript could be considered for publication in *Nat. Commun.*

1. Why decreasing the amount of 2,6-lutidine led to a significantly improved yield? What are the mechanistic implications of this observation?
2. Why does changing the ethyl group to the methyl group have such a significant impact on the reaction yield (3b vs 3r)?
3. What were the d.r. for products 3u and 3v? If they are the single diastereomer, why do they give such an excellent selectivity?
4. Why Ir-based catalyst is better than Ru-based catalyst when unactivated alkenes were used?
5. The authors state that "a tri-substituted alkene of cholesterol reacted cleanly to provide the corresponding product 5n in an unoptimized 21% yield and with excellent diastereo- (> 20:1 dr) and regioselectivity." what happened to the rest of the starting materials?
6. How well does the synthesis of *a,b*-unsaturated esters work for unactivated alkene substrates (Table 6)?
7. A radical clock experiment was included in Fig. 3, but it was not discussed in the main text.
8. The authors should perform quenching experiments and quantum yield measurements to support the proposed reaction mechanism.
9. Typos: (i) page 7: "As shown in Table 3, by treatment of *N*-arylacrylamides 7..." it should be Table 8; (ii) page 8, "a single electron transfer from *Ir(ppy)₃ to compound 2a" should be "compound 2a".
10. The sentence (page 8) "In the meantime, a single electron transfer from *Ir(ppy)₃" is a little bit confusing or redundant because it has been mentioned in the earlier sentence. "Ethyl chlorooxalacetate (2a) would accept an electron from *Ir(ppy)₃". The authors should write it more concisely.
11. The Subtitle "Discussion" should be "Conclusion."

Reviewer #2 (Remarks to the Author):

In this work Prof. Jie Wu and coworkers developed a difunctionalization of alkenes in presence of alkyloxalyl chlorides under photocatalytic conditions. The reaction is promoted by a highly efficient reductive photocatalyst able to perform the SET reduction of the alkyloxalyl chloride compound. This after a fast decarbonylation generates an alkoxyacylation radical prone to perform a Giese type reaction with a double bond. Unfortunately, the presence of anion chloride closes the cycle with the formation of a β -chloro ester. The main advantage of the protocol would be the synthesis of the alkyloxalyl chloride in-situ, but no pictures of the experimental procedure (photoreactor, flask reaction...) are reported. The paper is lightly misleading. Indeed, it is presented as an alkoxyacylation reaction, even this happening just for a restricted number of substrates.

The literature about generation of esters from alkenes is well documented with different organometallic strategies but barely reported in photocatalysis. The most influential paper in this field (Org. Lett., 2016, 18, 2564-2567) allow the functionalization of alkenes with phthalamide derivatives. In this context, the present paper cannot achieve a so similar structures without the introduction of further reactions. The authors suggest that their reaction avoid the use of CO gas and high temperature, but it is not completely true. Indeed, CO gas is produced as side product and temperatures around 80 °C are needed.

Moreover, the mechanism explanation is not sufficient, especially due to the absence of such experiments as Stern-Volmer and CV of substrates.

In conclusion, the global idea does not achieve the standards of a journal as Nature Communications.

Reviewer #3 (Remarks to the Author):

In the ms presented by Prof. Wu and coworkers an interesting photocatalyzed alkoxy-carbonylation of alkenes is reported. As mentioned in the introduction section, the direct incorporation of a COOR group in an organic compound is a challenge and only a few examples are known by using the photochemical tool. In this respect, the Authors may include a review on this topic (10.1039/C8GC03810D) and should compare the recent similar 1,2-methoxy methoxycarbonylation of alkenes (<https://dx.doi.org/10.1021/acscatal.0c04332>) with your work.

However, the Authors found a tricky way to form an alkoxy-carbonyl radical starting from alkyloxalyl chlorides. The process is efficient since several styrenes along with other substituted alkenes were easily functionalized. The versatility of the method is witnessed by the fact that it is possible to tune the reaction conditions in order to shift from an alkoxy-carbonylchlorination to the synthesis of α,β -unsaturated esters or other reactions.

The number of examples presented is impressive and varied and the mechanism reported is plausible. Accordingly, I suggest the publication of this work after the clarification of a couple of points as detailed in the following.

- 1) I wonder why the decarbonylation is so efficient despite the low stability of the alkoxy-carbonyl radical formed. I noticed that in most cases the reaction is carried out at 60 °C. It is not clear which is the effect of the temperature in the reaction (and especially on the decarbonylation step). Have the Authors observed the trapping of the carbonylated radical onto the C=C bond? In the future, to avoid thermal contributions the Authors may take care of the temperature taking it lower than 40 °C.
- 2) To complete the work, the measurements of the reduction potential of the radical precursors should be added to the work in order to strengthen the mechanism proposed.

The detailed corrections are listed below point by point:

1. Page 1, title: "Efficient Access to Aliphatic Esters by Photocatalyzed Alkoxyacylation of Alkenes" was changed to "Efficient access to aliphatic esters by photocatalyzed alkoxyacylation of alkenes with alkyloxalyl chlorides".
2. Page 1, authors: "Jian-Qiang Chen,^{*},¹ Xiaodong Tu,¹ Qi Tang,¹ Ke Li,¹ Liang Xu,¹ Siyu Wang,¹ Mingjuan Ji¹ and Jie Wu^{*,1,2,3}" was changed to "Jian-Qiang Chen^{1*}, Xiaodong Tu¹, Qi Tang¹, Ke Li¹, Liang Xu¹, Siyu Wang¹, Mingjuan Ji¹ & Jie Wu^{1,2,3*}".
3. Page 1, added: "to J.-Q.C. (email: chenjq@tzc.edu.cn) or"
4. Page 2, right column, paragraph 2: "generated most commonly" was changed to "generated most".
5. Page 3, left column, paragraph 1, added: "Results" and "Reaction optimization." .
6. Page 3, right column, "Table 2" was corrected to "Fig. 2".
7. Page 3, right column, Fig. 2 footnote, added: "The dr values were determined by ¹H NMR analysis. ^a Determined by HPLC analysis."
8. Page 3, right column: "increased" was changed to "higher".
9. Page 4, left column, paragraph 1, added: "Substrate scope of activated alkenes."
10. Page 4, left column, paragraph 3, added: "Reaction conditions development."
11. Page 4, left column, paragraph 3, added: "In general, reactions of secondary alkyl chlorooxoacetates gave better yields than those of primary alkyl chlorooxoacetates. This result could be rationalized by the slower decay rate of secondary alkoxyacyl radicals ($\tau = 4.4 \mu\text{s}$ for *i*-propylcarbonyloxy radical *vs.* $\tau = 2.4 \mu\text{s}$ for ethylcarbonyloxy radical).²²"
12. Page 4, right column, "Table 3" was changed to "Table 2".
13. Page 4, right column, "Table 4" was changed to "Fig. 3".
14. Page 4, right column, Table 4: "^bReaction time: 84 h." was changed to "^a Reaction time: 84 h. The dr value was determined by ¹H NMR analysis."
15. Page 5, left column, paragraph 1, added: "Substrate scope of unactivated alkenes."
16. Page 5, left column, paragraph 1: "even in cases where benzyl, cyclohexyl and *tert*-butyl were present in the *a*-position relative to the double bonds." was changed to "even in cases where benzyl, cyclohexyl and *tert*-butyl were present in the *a*-position of double bond."
17. Page 5, left column, "Table 5" was changed to "Fig. 4".
18. Page 5, right column, "Table 6" was changed to "Fig. 5".
19. Page 5, right column, paragraph 1, added: "Synthesis of α,β -unsaturated esters."
20. Page 5, right column, paragraph 1: "This formal β -selective alkenyl C-H alkoxyacylation strategy is quite appealing." was changed to "we envisioned that this formal β -selective alkenyl C-H alkoxyacylation could be accomplished and quite appealing."
21. Page 6, "Table 7" was changed to "Fig. 6".
22. Page 6, footnote, added: "The dr values were determined by ¹H NMR analysis."
23. Page 6, footnote: "^b Reaction time: 48 h." was changed to "^a Reaction time: 48 h."
24. Page 7, left column, paragraph 1, added: "Synthesis of oxindole-3-acetates."
25. Page 7, left column, paragraph 2, added: "We speculated that the unique three-dimensional structure of β -cholestanol might influence the following cyclization reaction."
26. Page 7, right column, paragraph 1: "However, transformations of benzyl oxalyl monochloride and tertiary alcohol derivative were not successful." was corrected to

- “However, reactions of benzyl and tertiary alcohol derivatives failed to provide the corresponding products.”
27. Page 7, right column, “Table 8” was changed to “Fig. 7”.
 28. Page 9, right column, paragraph 1, added: “Synthesis of furoindolines”.
 29. Page 7, right column, added: “Synthetic application”.
 30. Page 7, right column, Fig. 8: “Fig. 2 Further applications” was changed to “Fig. 8 Synthetic application. **a** Synthesis of expensive dihydronaphthalene derivative. **b** Formal synthesis of marketed drug ozagrel. **c** Synthesis of 1,4-dicarbonyl compound.”.
 31. Page 8, Fig. 8: “6o, 6o” was changed to “6q, 6q”.
 32. Page 8, left column, added: “Mechanistic studies”.
 33. Page 8, “Fig. 3 Control experiment and proposed reaction mechanism” was changed to “Fig. 9 Mechanistic studies. **a** Radical clock experiment. **b** Cyclic voltammetry studies. **c** Luminescence quenching experiment. **d** Cl⁻ and pyridinium salt (I-2a) enhance the product formation. **e** Plausible mechanism.”.
 34. Page 10, right column, Methods: “General procedure for the alkoxy carbonyl chlorination of activated alkenes:” was changed to “General procedure for the synthesis of compound **3**”.
 35. Page 10, Methods, added: “General procedures for the synthesis of compounds **5** and **5'**; General procedures for the synthesis of compound **6**; General procedures for the synthesis of compound **8**; General procedures for the synthesis of compound **9**”.
 36. Manuscript: All the changes are shown in the manuscript text file with color highlighting. (Manuscript_track changes).
 37. Supporting Information: All the changes are shown in the supporting information text file with color highlighting. (Supporting Information_track changes).

Response to Reviewer 1:

Question 1: This paper is missing (i) many explanations and rationales for different observations. In addition, (ii) mechanistic studies and understandings.

√ Thanks for the suggestion. Page 4, left column, paragraph 3, added: “In general, reactions of secondary alkyl chlorooxoacetates gave better yields than those of primary alkyl chlorooxoacetates. This result could be rationalized by the slower decay rate of secondary alkoxy carbonyl radicals ($\tau = 4.4 \mu\text{s}$ for *i*-propylcarbonyloxy radical *vs.* $\tau = 2.4 \mu\text{s}$ for ethylcarbonyloxy radical).²²”

Page 7, left column, paragraph 2, added: “We speculated that the unique three-dimensional structure of β -cholestanol might influence the following cyclization reaction.”

TEMPO trapping experiment, ¹H NMR studies, cyclic voltammetry studies, Stern-Volmer fluorescence quenching experiments and quantum yield measurements were performed and the results were provided in this manuscript. For details, please see the text and Supporting Information.

Question 2: Why decreasing the amount of 2,6-lutidine led to a significantly improved yield? What are the mechanistic implications of this observation?

√ Thanks for the suggestion. As shown in Figure 9, a series of experiments were performed to understand the role of 2,6-lutidine. As mentioned above, both photosensitizer and visible light were necessary for this alkoxy carbonylation process (Table 1, entries 9 and 10). Thus, these results could rule out the ethyl chlorooxoacetate activation by the formation of electron-donor-acceptor (EDA) complex. The presence of 2,6-lutidine enhanced the

reduction potential of ethyl chlorooxoacetate **2a** from $E_p(\mathbf{2a}) = -1.23 \text{ V vs. Ag/AgCl}$ to $E_p(\mathbf{2a}+2,6\text{-lutidine}) = -1.09 \text{ V vs. Ag/AgCl}$. $^1\text{H NMR}$ studies showed a critical downfield shift of 2,6-lutidine protons after the addition of ethyl chlorooxoacetate **2a** (for details, see Figure S4), indicating the generation of acyl pyridinium salt **I-2a**. The possible structure of **I-2a** was shown in Figure 9. These results implied that there was no π conjugation between the aromatic ring and acyl plane. Stern-Volmer experiments revealed that ethyl chlorooxoacetate **2a** and acyl pyridinium salt **I-2a** could quench the excited photocatalyst, but alkene and 2,6-lutidine could not. The quenching constant for $^*\text{Ir}(\text{ppy})_3$ with ethyl chlorooxoacetate **2a** was determined as $k_q = 1.70 \times 10^8 \text{ M}^{-1}\text{s}^{-1}$, which was 1.38 times faster than the reaction of $^*\text{Ir}(\text{ppy})_3$ with **I-2a**.

Page 9, column 1, last paragraph, added: “It is noteworthy that the presence of 2,6-lutidine was significant for the success of these transformations. As mentioned above, both photosensitizer and visible light were necessary for this alkoxyacylation process (Table 1, entries 9 and 10). Thus, these results could rule out the ethyl chlorooxoacetate activation by the formation of electron-donor-acceptor (EDA) complex.⁵⁵ As shown in Figure 9b, the presence of 2,6-lutidine enhanced the reduction potential of ethyl chlorooxoacetate **2a** from $E_p(\mathbf{2a}) = -1.23 \text{ V vs. Ag/AgCl}$ to $E_p(\mathbf{2a}+2,6\text{-lutidine}) = -1.09 \text{ V vs. Ag/AgCl}$. $^1\text{H NMR}$ studies showed a critical downfield shift of 2,6-lutidine protons after the addition of ethyl chlorooxoacetate **2a** (for details, see Supporting Information), indicating the generation of acyl pyridinium salt **I-2a**. The possible structure of **I-2a** was shown in Figure 9. These results implied that there was no π conjugation between the aromatic ring and acyl plane. Stern-Volmer experiments revealed that ethyl chlorooxoacetate **2a** and acyl pyridinium salt **I-2a** could quench the excited photocatalyst, but alkene and 2,6-lutidine could not. The quenching constant for $^*\text{Ir}(\text{ppy})_3$ with ethyl chlorooxoacetate **2a** was determined as $k_q = 1.70 \times 10^8 \text{ M}^{-1}\text{s}^{-1}$, which was 1.38 times faster than the reaction of $^*\text{Ir}(\text{ppy})_3$ with **I-2a**.”

Question 3: Why does changing the ethyl group to the methyl group have such a significant impact on the reaction yield (3b vs 3r)?

√ Thanks. We speculated that this result could be rationalized by the slower decay rate of ethylcarbonyloxy radical than methylcarbonyloxy radical (from *Eur. J. Org. Chem.* **2001**, 545–552).

Question 4: What were the d.r. for products **3u** and **3v**? If they are the single diastereomer, why do they give such an excellent selectivity?

√ The dr values of compound **3u** and **3v** were determined by HPLC analysis, and the results were provided in Fig. 2 and Supporting Information.

Question 5: Why Ir-based catalyst is better than Ru-based catalyst when unactivated alkenes were used?

√ As shown in Supporting Information (Table S12 vs. S14), the quenching constant for $^*\text{Ir}(\text{ppy})_3$ with ethyl chlorooxoacetate **2a** was determined as $k_q = 1.70 \times 10^8 \text{ M}^{-1}\text{s}^{-1}$, which was 110 times faster than the reaction of $^*\text{Ru}(\text{bpy})_3\text{Cl}_2$ with **2a** ($k_q = 1.54 \times 10^6 \text{ M}^{-1}\text{s}^{-1}$). Additionally, [2+2] cycloaddition product could be found when Ir-based catalysts were used in this transformation. In 2018, Wu's group reported a similar [2+2] cycloaddition of aryl terminal olefins by using high triplet-energy $\text{Ir}(\text{ppy})_3$ as the photocatalyst. $\text{Ru}(\text{bpy})_3\text{Cl}_2$ showed no catalytic activity for this [2+2] cycloaddition (*CCS Chem.* **2019**, 1, 582–588).

Question 6: The authors state that "a tri-substituted alkene of cholesterol reacted cleanly to provide the corresponding product **5n** in an unoptimized 21% yield and with excellent diastereo- (> 20:1 dr) and regioselectivity." what happened to the rest of the starting materials?

√ Thanks. The rest of starting material was recovered.

Page 5, column 1: "Excitingly, a tri-substituted alkene of cholesterol reacted cleanly to provide the corresponding product **5n** in an unoptimized 21% yield and with excellent diastereo- (> 20:1 dr) and regioselectivity." was changed to "Reaction of tri-substituted alkene derived from cholesterol produced the corresponding product **5n** in an unoptimized 21% yield, along with 40% of unreacted starting material recovered. Another aspect worth mentioning here was the excellent diastereo- (> 20:1 dr) and regioselectivity of this transformation."

Question 7: How well does the synthesis of α,β -unsaturated esters work for unactivated alkene substrates (Table 6)?

√ Thanks for the suggestion. "Table 6" was changed to "Fig. 5". Unactivated alkenes could be employed in the reaction, affording the desired products (**6o** and **6p**) in moderate yields. The results were provided in Fig. 5.

Question 8: A radical clock experiment was included in Fig. 3, but it was not discussed in the main text.

√ Thanks for the suggestion. "Fig. 3" has been corrected to "Fig. 9".

The discussion was provided in page 9, column 1, last paragraph, added: "To elucidate the possible reaction mechanism, a radical clock experiment with (1-cyclopropylvinyl)benzene **1x** as the substrate was preformed, and a ring-opening product was obtained as expected (Fig. 9a). This result clearly demonstrated the involvement of an alkoxy carbonyl radical."

Question 9: The authors should perform quenching experiments and quantum yield measurements to support the proposed reaction mechanism.

√ Thanks for the suggestion. Stern-Volmer fluorescence quenching experiments and quantum yield measurements were performed. The results were added in this text, and the details were provided in Supporting Information.

Page 9, column 2, added: "Stern-Volmer experiments revealed that ethyl chlorooxoacetate **2a** and acyl pyridinium salt **I-2a** could quench the excited photocatalyst, but alkene and 2,6-lutidine could not. The quenching constant for $^*Ir(ppy)_3$ with ethyl chlorooxoacetate **2a** was determined as $k_{q1} = 1.70 \times 10^8 \text{ M}^{-1}\text{s}^{-1}$, which was 1.38 times faster than the reaction of $^*Ir(ppy)_3$ with **I-2a**. We speculated that the unique three-dimensional structure of **I-2a** might influence the single electron transfer from the excited photocatalyst to **I-2a**. Notably, the addition of Cl^- would slightly improve the yield of **3a** from 3 to 20% (Fig. 9d). This result suggested the feasibility of carbon cation pathway. The quantum yields of these reactions were determined to be 1.11, 1.33 and 9.98, showing the extended radical-chain reactions were possible. Moreover, bond dissociation energy (BDE) of C-H bond could reflect the stability of carbon radicals. The C-H bond dissociation energy of $MeOC(O)-H$ and ^tBu-H are 95.4⁵⁶ and 95.6 kcal mol⁻¹,⁵⁷ respectively, and the stability of alkoxy carbonyl radical might be similar to the tertiary carbon radical."

Question 10: Typos: (i) page 7: "As shown in Table 3, by treatment of N-arylacrylamides 7..." it should be Table 8; (ii) page 8, "a single electron transfer from $^*Ir(ppy)_3$ to compound 2a" should be "compound 2a".

√ Thanks. The mentioned typo errors were corrected. Additionally, the entire text was checked, and some errors were corrected.

Question 11: The sentence (page 8) "In the meantime, a single electron transfer from $^*Ir(ppy)_3$ " is a little bit confusing or redundant because it has been mentioned in the earlier sentence. "Ethyl chlorooxoacetate (**2a**) would accept an electron from $^*Ir(ppy)_3$ ". The authors should write it more concisely.

√ Thanks. The whole paragraph was rewritten.

Page 9, column 2, paragraph 2: "On the basis of the above experimental results and previous reports,⁵⁸ a plausible reaction mechanism is proposed in Figure 9e. Excitation of photosensitizer $Ir(ppy)_3$ **16** with blue light would generate a long-lived excited $^*Ir(ppy)_3$ ($\tau = 1.9 \mu s$).⁵⁹ Meanwhile, 2,6-lutidine would condense with ethyl chlorooxoacetate **2a** to form acyl pyridinium salt **I-2a**. The highly reducing species $^*Ir(ppy)_3$ **17** ($E_{1/2} [Ir(ppy)_3^+ /] = -1.73$ V vs. SCE) would reduce intermediate **I-2a** leading to acyl radical intermediate **12**, 2,6-lutidine and $Ir(ppy)_3^+$ **18**. Acyl radical **12** would then undergo decarbonylation, giving rise to alkoxy carbonyl radical **13**.^{60,61} The driving force for this transformation is the extrusion of carbon monoxide and the generation of stable alkoxy carbonyl radical. This ambiphilic radical **13** could react with an alkene to afford alkyl radical **14**, which would be readily oxidized by $Ir(ppy)_3^+$ **18** to regenerate ground-state photocatalyst **16** and carbocation intermediate **15**. This carbocation intermediate **15** would be attacked by chloride anion leading to the desirable β -chloro ester **5** (path A). A competitive chain pathway could not be excluded (path B). Alkyl radical **14** would abstract chlorine atom from ethyl chlorooxoacetate **2a** to produce the target product **5** and regenerate acyl radical intermediate **12**."

Question 12: The Subtitle "Discussion" should be "Conclusion."

√ Thanks. The subtitle "Discussion" was corrected to "Conclusion".

Response to Reviewer 2:

Question 1: No pictures of the experimental procedure (photoreactor, flask reaction...) are reported.

√ Thanks for the suggestion. The pictures of experimental devices were provided in the Supporting Information.

Question 2: The paper is lightly misleading. Indeed, it is presented as an alkoxy carbonylation reaction, even this happening just for a restricted number of substrates.

√ Thanks for the suggestion. Page 1, title: "Efficient Access to Aliphatic Esters by Photocatalyzed Alkoxy carbonylation of Alkenes" was changed to "Efficient access to aliphatic esters by photocatalyzed alkoxy carbonylation of alkenes with alkyloxalyl chlorides".

Question 3: In this context, the present paper cannot achieve a so similar structures without the introduction of further reactions.

√ Thanks for the suggestion. As shown in Fig. 8c, coupling of ethyl chlorooxoacetate **2a** (3.0 equiv) with benzyl acrylate in the presence of 2 mol % of $Ir(ppy)_3$, 2.0 equiv of diethyl

1,4-dihydro-2,6-dimethyl-3,5-pyridinedicarboxylate in MeCN with irradiation at 30 °C with blue LEDs gave 1,4-dicarbonyl compound **5d'** in 85% yield.

Page 9, column 1, paragraph 2, added: “Overman demonstrated that methoxycarbonyl radical could react with electron-deficient olefins to afford 1,4-dicarbonyl compounds.¹³ We next applied our methodology to this transformation. As expected, coupling of ethyl chlorooxoacetate **2a** (3.0 equiv) with benzyl acrylate in the presence of Ir(ppy)₃ (2 mol %) and diethyl 1,4-dihydro-2,6-dimethyl-3,5-pyridinedicarboxylate (2.0 equiv) in MeCN with blue LEDs irradiation at 30 °C gave rise to 1,4-dicarbonyl compound **5d'** in 85% yield (Fig. 8c).”

Question 4: The authors suggest that their reaction avoid the use of CO gas and high temperature, but it is not completely true. Indeed, CO gas is produced as side product and temperatures around 80 °C are needed.

√ Thanks for the suggestion. Page 2, left column, paragraph 1: “these processes rely on the use of high pressure of toxic and flammable CO at high temperature, which often require specific equipment and safety precautions.” was changed to “these processes rely on the use of high pressure of CO, which often require specific equipment and safety precautions.”

Question 5: Moreover, the mechanism explanation is not sufficient, especially due to the absence of such experiments as stern-volmer and CV of substrates.

√ Thanks for the suggestion. Stern-Volmer fluorescence quenching experiments and cyclic voltammetry studies were performed, and the results were added in the text (page 9) and Supporting Information.

Response to Reviewer 3:

Question 1: In this respect, the Authors may include a review on this topic (10.1039/C8GC03810D) and should compare the recent similar 1,2-methoxy methoxycarbonylation of alkenes (<https://dx.doi.org/10.1021/acscatal.0c04332>) with your work.

√ Thanks for the suggestion. The mentioned reference (10.1039/C8GC03810D) was added as ref 17: Raviola, C., Protti, S., Ravelli, D., Fagnoni, M. Photogenerated acyl/alkoxycarbonyl/carbamoyl radicals for sustainable synthesis. *Green Chem.* **21**, 748–764 (2019).

Page 2, column 2, paragraph 1: “Additionally, alkoxycarbonyl radicals can be formed from carbazates by treatment with metal catalysts and stoichiometric quantities of oxidants.¹¹” was changed to “Additionally, alkoxycarbonyl radicals can be formed from carbazates¹¹ and alkyl formates¹² by treatment with stoichiometric amounts of oxidants.”. Moreover, this transformation was added to Fig. 1c.

The mentioned reference (10.1021/acscatal.0c04332) was provided as ref. 12 (Zheng, M., Hou, J., Zhan, L.-W., Huang, Y., Chen, L., Hua, L.-L., Li, Y., Tang, W.-Y., Li, B.-D. Visible-light-driven, metal-free divergent difunctionalization of alkenes using alkyl formates. *ACS Catal.* **11**, 542–553 (2021).). In this paper (10.1021/acscatal.0c04332), most of the reported examples dealt with structurally simple alkoxycarbonyl radicals.

Page 2, column 2, paragraph 1, to compare the similar 1,2-methoxy methoxycarbonylation of alkenes (ref. 12, <https://dx.doi.org/10.1021/acscatal.0c04332>), added: “Moreover, most of the reported examples dealt with structurally simple alkoxycarbonyl radicals,¹² while only a few reports exploiting complex alkoxycarbonyl radicals was described.¹⁴”

Question 2: I wonder why the decarbonylation is so efficient despite the low stability of the alkoxy carbonyl radical formed. I noticed that in most cases the reaction is carried out at 60 °C. It is not clear which is the effect of the temperature in the reaction (and especially on the decarbonylation step). Have the Authors observed the trapping of the carbonylated radical onto the C=C bond? In the future, to avoid thermal contributions the Authors may take care of the temperature taking it lower than 40 °C.

√ Thanks for the suggestion. Bond dissociation energy (BDE) of C–H bond can reflect the stability of carbon radicals. The C–H Bond dissociation energy of MeOC(O)–H and ^tBu–H are 95.4, and 95.6 kcal mol⁻¹, respectively. The stability of alkoxy carbonyl radical may be similar to the tertiary carbon radical.

BDE of C-H bonds (kcal·mol⁻¹)

the order of radical stability

As shown in below, we investigated this reaction under lower temperature. As a result, the yield of compound **8a** was decreased and we did not observe the formation of any of the carbonylation product **8a'**.

Question 3: To complete the work, the measurements of the reduction potential of the radical precursors should be added to the work in order to strengthen the mechanism proposed.

√ Thanks for the suggestion. As shown in Fig. 9, the reduction potential of ethyl chlorooxoacetate **2a** was measured by the cyclic voltammetry.

REVIEWER COMMENTS

Reviewer #1 (Remarks to the Author):

The authors addressed most of my questions. Some questions, however, remain unanswered and inconsistent with the proposed mechanism. First, the author described that the formation of I-2a results in a more positive reduction potential, but it quenches the excited photocatalyst less efficiently. In such a case, does the formation of I-2a truly benefit the reaction? In addition, based on the proposed mechanism, a catalytic amount of 2,6-lutidine should be enough, but the reaction requires 1.5 equiv of 2,6-lutidine. Is the 2,6-lutidine really play the role that the authors proposed? Furthermore, the authors claimed that ethylcarbonyloxy radical decays slower than methylcarbonyloxy radical and "supported" their statement with *Eur. J. Org. Chem.* 2001, 545. However, there is no methylcarbonyloxy example in the *Eur. J. Org. Chem.* paper. In fact, the *Eur. J. Org. Chem.* paper is speaking against the authors' claims. If one considers the decay rate, ethylcarbonyloxy radical should decay faster than methylcarbonyloxy radical because the resulting ethyl radical is more stable than the methyl radical. Thus, the authors need to address the inconsistencies described above before publishing this manuscript.

Reviewer #2 (Remarks to the Author):

In this revision, most of the technical problems of the manuscript have been corrected, but I still think, that this work is not enough original for a journal such as *Nature Communications*. In the literature, it can be found the generation of esters from with different organometallic strategies. Even, the most influential paper in this field (*Org. Lett.*, 2016, 18, 2564-2567) allow the functionalization of alkenes with phthalamide ester derivatives. Therefore, this is a slightly variation for known protocols and the global idea does not achieve the standards of a journal as *Nature Communications*.

Reviewer #3 (Remarks to the Author):

In the revised version of the ms, Prof Wu and coworkers addressed all of the issues I have raised in my former report. The ms now could be acceptable in the present form.

Response to Reviewer 1:

Question 1: First, the author described that the formation of **I-2a** results in a more positive reduction potential, but it quenches the excited photocatalyst less efficiently. In such a case, does the formation of **I-2a** truly benefit the reaction?

Free-energy reaction profiles (kcal/mol) calculated with the SMD (acetonitrile) M062X/6-31+G(d) level at 303 K.

√ Thanks for the suggestion. In order to shed more light on the reaction mechanism, especially the role of 2,6-lutidine, DFT calculations were carried out with the Gaussian 09 software package. The calculation details were provided in the Supporting Information. As shown in Figure 9e, the whole reaction was thermodynamically favorable and the driving force for this transformation was the extrusion of carbon monoxide, the generation of stable alkoxy carbonyl radical **13** and subsequent reaction with alkene to afford alkyl radical **14**. Concerning the formation of key intermediate **12**, though the generation of acyl pyridinium salt **I-2a** was a little bit endothermic, the formation of **I-2a'** via reduction of **I-2a** by highly reducing species *Ir(ppy)_3 was more thermodynamically favored compared to the direct formation of **19** without partition of 2,6-lutidine. Additionally, anion radical **19** could react with compound **2a** to deliver the undesired alkyl radical **20**. Thus 2,6-lutidine might facilitate the transformation relative to no participation of 2,6-lutidine.

Page 9, column 2, paragraph 2, added: "In order to shed more light on the reaction mechanism, especially the role of 2,6-lutidine, DFT calculations were carried out with the Gaussian 09 software package.⁵⁸⁻⁶⁰ The calculation details were provided in the Supporting Information. As shown in Figure 9e, the whole reaction was thermodynamically favorable and the driving force for this transformation was the extrusion of carbon monoxide, the generation of stable alkoxy carbonyl radical **13** and subsequent reaction with alkene to afford alkyl radical **14**. Concerning the formation of key intermediate **12**, though the generation of acyl pyridinium salt **I-2a** was a little bit endothermic, the formation of **I-2a'** via reduction of **I-2a** by highly reducing species *Ir(ppy)_3 was more thermodynamically favored compared to the direct formation of **19** without partition of 2,6-lutidine. Additionally, anion radical **19**

could react with compound **2a** to deliver the undesired alkyl radical **20**. Thus 2,6-lutidine might facilitate the transformation relative to no participation of 2,6-lutidine."

Question 2: In addition, based on the proposed mechanism, a catalytic amount of 2,6-lutidine should be enough, but the reaction requires 1.5 equiv of 2,6-lutidine. Is the 2,6-lutidine really play the role that the authors proposed?

✓ Thanks for the suggestion. The formation of **I-2a'** *via* the reduction of **I-2a** by highly reducing species $^*Ir(ppy)_3$ was more thermodynamically favored compared to the direct formation of **19** without participation of 2,6-lutidine. Stern-Volmer experiments have demonstrated that the step to generate anion radical **19** is kinetically favorable. Hence, raising the equivalent of 2,6-lutidine would facilitate the generation of the **I-2a'** and suppress the formation of anion radical **19**.

Page 9, right column, paragraph 2, added: “Moreover, Stern-Volmer experiments have demonstrated that the step to generate anion radical **19** is kinetically favorable. Hence, raising the equivalent of 2,6-lutidine would facilitate the generation of the **I-2a'** and suppress the formation of anion radical **19**.”

Question 3: The authors claimed that ethylcarbonyloxy radical decays slower than methylcarbonyloxy radical and "supported" their statement with Eur. J. Org. Chem. 2001, 545. However, there is no methylcarbonyloxy example in the Eur. J. Org. Chem. paper. In fact, the Eur. J. Org. Chem. paper is speaking against the authors' claims. If one considers the decay rate, ethylcarbonyloxy radical should decay faster than methylcarbonyloxy radical because the resulting ethyl radical is more stable than the methyl radical. Thus, the authors need to address the inconsistencies described above before publishing this manuscript.

✓ Thanks for the suggestion. “The alkoxy carbonyl radicals ($\nu_{C=O} = 1802\text{ cm}^{-1}$ for R = ethyl) generally have a lifetime of several microseconds, decaying by reaction with the

solvent (CCl₄) to yield esters of chloroformic acid.” (From: *Eur. J. Org. Chem.* **2001**, 545-552, Abstract)

As shown in Fig. S15, DFT calculations were also applied to study the side reaction of ethoxycarbonyl radical and ethyl chlorooxoacetate to generate radical **12** and ethyl carbonochloridate. The reaction free energy was 0.6 kcal/mol, while the reaction free energy of methoxycarbonyl radical and methyloxalyl chloride was only 0.1 kcal/mol. According to Arrhenius equation, the side reaction of the latter was more than the former. This might contribute to the low yield of methyloxalyl chloride compared to ethyloxalyl chloride (**3b** vs **3r**). By raising the equivalent of methyl oxalyl chloride, the yield of **3r** was increased from 38% to 54%.

Page 9, right column, added: “As shown in Fig. S15, DFT calculations were also applied to study the side reaction of ethoxycarbonyl radical and ethyl chlorooxoacetate to generate radical **12** and ethyl carbonochloridate. The reaction free energy was 0.6 kcal/mol, while the reaction free energy of methoxycarbonyl radical and methyloxalyl chloride was only 0.1 kcal/mol. According to Arrhenius equation, the side reaction of the latter was more than the former. This might contribute to the low yield of methyloxalyl chloride compared to ethyloxalyl chloride (**3b** vs **3r**).”.

Response to Reviewer 2:

Question 1: In this revision, most of the technical problems of the manuscript have been corrected, but I have still think, that this works is not enough original for a journal such as Nature Communications. In the literature, it can be found the generation of esters from with different organometallic strategies. Even, the most influent paper in this field (*Org. Lett.*, 2016, 18, 2564-2567) allow the functionalization of alkenes with phtalamide ester derivatives. Therefore, this is a slightly variation for known protocols and the global idea does not achieve the standards of a journal as Nature Communications.

√ Thanks. It was reported that alkoxy carbonyl radicals could be produced by photoredox-catalyzed fragmentation of methyl *N*-phthalimidoyl oxalates (Fig. 1c). These existing methods for generating alkoxy carbonyl radicals from alcohols require multistep synthetic procedures. Moreover, most of the reported examples dealt with structurally simple alkoxy carbonyl radicals. In this report, a new strategy to access aliphatic esters from olefins through a unique photocatalyzed alkoxy carbonylation reaction is described. Alkyloxalyl chlorides, generated in situ from the corresponding alcohols and oxalyl chloride, are engaged for the first time as alkoxy carbonyl radical fragments under photoredox catalysis. This transformation tolerates a broad scope of electron-rich and electron-deficient olefins and provides the corresponding β -chloro esters in good yields. Additionally, a formal β -selective alkene alkoxy carbonylation is developed. Moreover, a variety of oxindole-3-acetates and fuoroindolines are prepared in good to excellent yields. A more concise formal synthesis of

(±)-physovenine is accomplished as well. With these strategies, a wide range of natural-product-derived olefins and alkyloxalyl chlorides are also successfully employed.

Thanks for your consideration, and looking forward to your kind response.

Sincerely

Jie Wu

REVIEWERS' COMMENTS

Reviewer #1 (Remarks to the Author):

The authors have addressed all my concerns and the manuscript can now be accepted for publication.

Response to Reviewer 1:

Question: The authors have addressed all my concerns and the manuscript can now be accepted for publication.

√ Thanks for the suggestion.